Development and evaluation of a rapid visual loop-mediated isothermal amplification assay for the tcdA gene in Clostridioides difficile detection

Lin Minyi 1
Wang Pu 2
Lu Bingyun 3
Jin Ming 3
Tan Jiasheng 4
Liu Wei 5
Yuan Jing 5
Peng Xiaomou 1 pengxmou@mail.sysu.edu.cn
Chen Ye 3 yechen_fimmu@163.com
1 Department of Infectious Diseases, The Fifth Affiliated Hospital, Sun Yat-Sen University , Zhuhai , China
2 Department of Gastroenterology, Guangdong Provincial Key Laboratory of Gastroenterology, State Key Laboratory of Organ Failure Research, Nanfang Hospital, Southern Medical University , Guangzhou , China
3 Integrative Microecology Center, Shenzhen Key Laboratory of Gastrointestinal Microbiota and Disease, Shenzhen Clinical Research Center for Digestive Disease, Shenzhen Technology Research Center of Gut Microbiota Transplantation, Shenzhen Hospital, Southern Medical University , Shenzhen , China
4 Department of Gastroenterology, SongShan Lake Central Hospital of Dongguan City , Dongguan, Guangdong , China
5 Institute of Disease Control and Prevention, Academy of Military Medical Sciences , Beijing , China
LaMontagne Michael
Electronic publication date: 2024 Aug 30
Publication date: 2024
Volume: 12
Electronic Location ID: e17776
Received 2023 Nov 28; Accepted 2024 Jun 28
Copyright: © 2024 Lin et al.
Copyright year: 2024
Copyright holder: Lin et al.
License: This is an open access article distributed under the terms of the Creative Commons Attribution License, which permits unrestricted use, distribution, reproduction and adaptation in any medium and for any purpose provided that it is properly attributed. For attribution, the original author(s), title, publication source (PeerJ) and either DOI or URL of the article must be cited.
License URL: https://creativecommons.org/licenses/by/4.0/

Keywords: tcdA gene, Clostridioides difficile, Loop-mediated isothermal amplification, Visual detection

Funding: National Natural Science Foundation of China 82270581 GuangDong Basic and Applied Basic Research Foundation 2022A1515220213 National Key Research and Development Program of China 2021YFA0717001 Shenzhen Science and Technology Plan Project KCXFZ20211020163558024 This work was supported by the National Natural Science Foundation of China (grant number 82270581), the GuangDong Basic and Applied Basic Research Foundation (grant number 2022A1515220213), the National Key Research and Development Program of China (grant number 2021YFA0717001), and the Shenzhen Science and Technology Plan Project (grant number KCXFZ20211020163558024). The funders had no role in study design, data collection and analysis, decision to publish, or preparation of the manuscript.

==============================
Background

The tcdA gene codes for an important toxin produced by Clostridioides difficile (C. difficile), but there is currently no simple and cost-effective method of detecting it. This article establishes and validates a rapid and visual loop-mediated isothermal amplification (LAMP) assay for the detection of the tcdA gene.

Methods

Three sets of primers were designed and optimized to amplify the tcdA gene in C. difficile using a LAMP assay. To evaluate the specificity of the LAMP assay, C. difficile VPI10463 was used as a positive control, while 26 pathogenic bacterial strains lacking the tcdA gene and distilled water were utilized as negative controls. For sensitivity analysis, the LAMP assay was compared to PCR using ten-fold serial dilutions of DNA from C. difficile VPI10463, ranging from 207 ng/µl to 0.000207 pg/µl. The tcdA gene of C.difficile was detected in 164 stool specimens using both LAMP and polymerase chain reaction (PCR). Positive and negative results were distinguished using real-time monitoring of turbidity and chromogenic reaction.

Results

At a temperature of 66 °C, the target DNA was successfully amplified with a set of primers designated, and visualized within 60 min. Under the same conditions, the target DNA was not amplified with the tcdA12 primers for 26 pathogenic bacterial strains that do not carry the tcdA gene. The detection limit of LAMP was 20.700 pg/µl, which was 10 times more sensitive than that of conventional PCR. The detection rate of tcdA in 164 stool specimens using the LAMP method was 17% (28/164), significantly higher than the 10% (16/164) detection rate of the PCR method (X2 = 47, p < 0.01).

Conclusion

LAMP method is an effective technique for the rapid and visual detection of the tcdA gene of C. difficile, and shows potential advantages over PCR in terms of speed, simplicity, and sensitivity. The tcdA-LAMP assay is particularly suitable for medical diagnostic environments with limited resources and is a promising diagnostic strategy for the screening and detection of C. difficile infection in populations at high risk.

Introduction

Clostridioides difficile (C. difficile) is the major cause of antibiotic-associated diarrhea worldwide (Rodríguez et al., 2020). Although the estimated national burden of C. difficile infection (CDI) and associated hospitalizations decreased from 2011 through 2017 in 10 states in the US, the number of C. difficile cases in the US remained as high as 15,512 in 2017, with an estimated total national burden of 462,100 cases and an estimated incidence of 144 cases per 100,000 population (Guh et al., 2020). In the Shandong and Zhejiang provinces, China, tertiary hospitals reported a consistent 14% incidence of CDI (72/504) among hospitalized patients with suspected CDI in Shandong and 14% (115/804) among acute gastroenteritis outpatients in a Zhejiang pediatric hospital (Luo et al., 2018; Shuai et al., 2020). These findings are consistent with other studies from mainland China, which report a 14% crude incidence of toxigenic C. difficile in diarrheal patients (Tang et al., 2016).

The primary virulence factors of C. difficile are two structurally similar toxins-toxin A and toxin B, which are encoded by the tcdA and tcdB genes respectively (Kuehne, Cartman & Minton, 2011). Most cases of CDI are attributed to strains expressing both toxins A and B (A+B+) (Drudy, Fanning & Kyne, 2007). Although the toxin A−B+ strain is uncommon, it can also cause disease and has been relevant to previous outbreaks of CDI (Alfa et al., 2000; Kuijper et al., 2001; Drudy, Fanning & Kyne, 2007). Earlier reports indicated that the toxin A+B− strain only rarely caused human diseases (Rupnik, 2008). As a result, numerous studies have emphasized the importance of toxin B in the pathogenesis of CDI, while downplaying the importance of toxin A. However, a study recently reported the discovery of clinical pathogenic C. difficile strains that produce high levels of toxin A but minimal or no toxin B, indicating that toxin A alone can cause CDI (Lin et al., 2020). This pattern of toxin production, observed in more than 5% of isolates, is consistently found both in vitro and in vivo in humans and mice (Lin et al., 2020). Furthermore, the production of either toxin A or toxin B by these isolates is sufficient to induce the full spectrum of CDI symptoms (Drudy et al., 2007; Freeman et al., 2010). Additionally, both toxins A and B can independently cause disease in animal models (Kuehne, Cartman & Minton, 2011; Kuehne et al., 2010).

Current laboratory tests for the diagnosis of toxin A in C. difficile strains include the C.difficile cytotoxin neutralization assay (CCNA), toxigenic culture (TC), toxin A enzyme immunoassay (EIA), glutamate dehydrogenase (GDH) assay, and nucleic acid amplification test (NAAT).

Although CCNA and TC remain the current gold standards, their use for routine clinical detection is challenging due to their requirements for harsh culture conditions, involvement of highly technical and complex operations, and time-consuming nature (Shah et al., 2020; Liu et al., 2021). Enzyme immunoassays are specific and rapid, but not sensitive (Nicholson & Donskey, 2023). The glutamate dehydrogenase assay is sensitive and rapid, but it has some disadvantages such as cross-reactivity, poor specificity, and a high false positive rate (Bartlett, 2010; Crobach et al., 2016). Assays for the tcdA gene by NAAT include polymerase chain reaction (PCR; Kim et al., 2022b), multiplex-PCR (Moosavian et al., 2022), quantitative real-time PCR (Brennhofer et al., 2022), and multiplex real-time PCR (Novakova et al., 2021). Despite the specificity and sensitivity of these diagnostic methods, their suitability for rapid detection in primary hospitals and on-site detection is limited due to their time-consuming and complex nature, as well as the requirement for expensive equipment. Thus, a rapid, simple, and cost-effective assay is needed to complement current PCR methods for detecting the tcdA gene.

Loop-mediated isothermal amplification (LAMP) is a powerful molecular technique for nucleic acid amplification. LAMP leverages the strand displacement activity of Bst DNA polymerase, which facilitates DNA amplification under isothermal conditions (Notomi et al., 2000; Ushikubo, 2004). The high amplification efficiency of this technique, capable of generating up to 109 copies of target DNA within an hour, underscores its potential in rapid diagnostic applications. The excellent specificity of LAMP is attributed to its use of four (or six) primers, which can identify six (or eight) distinct regions on the target DNA or RNA (Notomi et al., 2000; Parida et al., 2008). Additionally, the detection limit of LAMP surpasses that of PCR, and the results can be visually interpreted without the need for sophisticated equipment. LAMP has been effectively used to detect various pathogens, including bacteria (Hong-Min et al., 2023), viruses (Nawab et al., 2024), parasites (Chen et al., 2023), and fungi (Badparva et al., 2022), as well as different toxin types (Norén et al., 2011; Pancholi et al., 2012), binary toxin genes (Yu et al., 2017), and resistant genes (Lin et al., 2015, 2022) of C.difficile.

This study designed three novel sets of LAMP primers and optimized LAMP for tcdA detection. To ascertain the specificity of the tcdA primer within the LAMP assay, 26 distinct pathogenic bacterial strains devoid of the tcdA gene were analyzed as negative controls. Primer sensitivity was assessed by conducting serial dilutions of C. difficile VPI10463 DNA. Finally, the study compared the consistency of LAMP and PCR methods in detecting the tcdA gene of C. difficile in 164 stool specimens.

Materials and Methods

Bacterial strains

A total of 26 pathogenic bacterial strains were selected to evaluate the specificity of the LAMP Assay (Table 1). C. difficile VPI10463 which carries the tcdA gene was used as the positive control. The tcdA gene of VPI10463 showed 100% identity with those of the tcdA gene in the sequence KC292122.1, which was confirmed by PCR-based sequencing (Fig. S1).

Table 1 Bacterial strains used in this study.

Species	Source	
C. difficile VPI10463	1	
Acinetobater baumannii H18	2	
Betahaemolytic streptococcus group A CMCC 32213	2	
Bordetella pertussis ATCC 18530	2	
Bacillus megatherium 4623	2	
Bacillus anthracis ATCC 9372	2	
Corynebacterium diphtheria CMCC 38001	2	
Enteropathogenic E. coli 2348	2	
Enterotoxigenic E. coli 44824	2	
Enteroinvasive E.coli 44825	2	
Mycobacterium tuberculosis 8362	2	
Neisseria meningitides group B CMCC29022	2	
Pseudomonas maltophilia ATCC13637	2	
Pseudomonas aeruginosa CMCC 10104	2	
Shigella flexneri 4536	2	
Shigella sonnei 2531	2	
Salmonella 10025819551001	2	
Salmonella paratyphosa 86423	2	
Salmonella aberdeen 9264	2	
Salmonella enteritidis 50326	2	
Staphylococcus aureus 2740	2	
Stenotrophomonas maltophilia H62	2	
Vibrio parahaemolyticus 5474	2	
Vibrio cholera 3802	2	
Vibrio carchariae 5732	2	
Yersinia enterocolitica 1836	2	
Yersinia pestis 2638	2	
Notes:

1 Lanzhou Institute of Microbiology.

2 Institute of Disease Control and Prevention, Academy of Military Medical Sciences.

Clinical stool specimens

Fresh stool specimens of suspected CDI inpatients with diarrhea were collected from August 1, 2013 to February 28, 2014 in Nanfang Hospital of Southern Medical University, Guangzhou, China. Inpatients over 18 years old who had received antibiotic or chemotherapy treatments within the past 60 days were included. Stool samples were included of patients who experienced diarrhea within 48 h of hospitalization, with no less than three episodes of diarrhea within a 24-h period, and with shapeless stool classified as Bristol types 5–7. The exclusion criteria for this study were as follows: patients who were under the age of 18; patients who had previously been sampled; patients with chronic diarrhea; patients who had used laxatives; patients with various types of infectious diarrhea, such as bacillary dysentery, typhoid fever, food poisoning, and amebic dysentery; patients with intestinal functional diseases, such as irritable bowel syndrome; patients with other types of diarrhea with clear causes unrelated to antibiotics, such as lactose intolerance; and patient samples that did not complete the entire testing process due to instrument or human errors. Of the 197 fresh stool specimens collected from inpatients with suspected CDI presenting diarrhea, 33 were excluded due to duplication (n = 23) or inadequate volume and freshness (n = 10). All stool specimens were frozen at 80 °C until detection. This study was performed in line with the principles of the Declaration of Helsinki. Approval was granted by the Ethics Committee of Fudan University Affiliated Huashan Hospital (committee ethic number: FDEC-2012-014). As a research member unit, Nanfang Hospital was successfully granted an ethics exemption by the Ethics Committee of Southern Medical University.

Informed consent statement

Because this study was based on the clinical examination of existing stool specimens forclinical research, it did not require patients to provide additional samples. In the clinical analysis, all patient identification information was expressed in code or pinyin to maintain the personal privacy of the patients and prevent individual health information from being exposed. The test results were only used for clinical research, and no test report was issued that would affect the diagnosis and treatment of the subjects. No patient risk was involved in this study. Therefore, an informed consent waiver was obtained from Nanfang Hospital of Southern Medical University for this study.

DNA extraction

To determine the specificity and sensitivity of the LAMP reactions under real conditions, genomic DNA was extracted from C. difficile VPI10463 and purified by the Wizard Genomic DNA Purification Kit (Promega, Madison, WI, USA). The purified genomic DNA was serially diluted in distilled water by a factor of 10, from 207 ng/ul to 0.000207 pg/ul. The concentration of pure genomic DNA before and after dilution was measured using the ND-1000 spectrophotometer (Thermo Fisher Scientific, Inc., Waltham, MA, USA). Genomic DNA of 26 pathogenic bacterial strains was extracted using a bacterial genomic DNA extraction kit (Tiangeng, Ningbo, China) according to the manufacturer’s instructions. Additionally, a stool genome extraction kit (Tiangeng, Ningbo, China) was employed to extract genomic DNA from stool samples. Genomic DNA was stored immediately at −20 °C until use.

Primer design

Based on the tcdA gene sequence of C. difficile obtained from NCBI GenBank database (GenBank accession number: X92982.1), three sets of LAMP primers were designed (Table 2). Further analysis of the sequences with the Primer Explorer V4 software (https://primerexplorer.jp/e/) yielded the outer forward primer (F3), outer backward primer (B3), forward inner primer (FIP), and backward inner primer (BIP). The FIP and BIP primers recognized both sense and antisense strands and were linked by a four-thymidine spacer (TTTT). The two loop primers (LF and LB) were designed to accelerate the amplification reaction. To compare the sensitivity and specificity of LAMP and PCR, conventional PCR was performed with the NK1 and NK2 primers (Table 2; Kato et al., 1991). All the primers were synthesized commercially (Sangon Biotech Co., Ltd., Shanghai, China).

Table 2 Primers used in LAMP and PCR.

Target gene	Primer	Type	Sequence (5′–3′)	
tcdA	tcdA-0F3	Forward outer	AGTTTGTTTACAGAACAAGAGTT	
	tcdA-0B3	Backward outer	ATCATTTCCCAACGGTCTA	
tcdA-0FIP	Forward inner	CCGCCAAAATTTTTTAGGGCTAATATTTATAGTCAGGAGTTGTTAAATCG	
tcdA-0BIP	Backward inner	AGATGTTGATATGCTTCCAGGTATTCCAATAGAGCTAGGTCTAGG	
tcdA-8F3	Forward outer	TCCAATACAAGCCCTGTAG	
tcdA-8B3	Backward outer	GAATCTCTTCCTCTAGTAGCT	
tcdA-8FIP	Forward inner	CTGCATTAATATCAGCCCATTGTTTTTGTATGGATAGGTGGAGAAGTC	
tcdA-8BIP	Backward inner	ACTGTGGTATGATAGTGAAGCATTCTTTCAGTGGTAGAAGATTCAACT	
tcdA-12F3	Forward outer	AGTTTGTTTACAGAACAAGAGTT	
tcdA-12B3	Backward outer	ATTTTATCATTTCCCAACGGT	
tcdA-12FIP	Forward inner	CCGCCAAAATTTTTTAGGGCTAATATTTTTATAGTCAGGAGTTGTTAAATCG	
tcdA-12BIP	Backward inner	AGATGTTGATATGCTTCCAGGTATTTTCTAGTCCAATAGAGCTAGGTC	
tcdA-12LF	Loop forward	CTTACTATGTCAGATGCTGCAGCTA	
tcdA-12LB	Loop backward	AGATGCTGCAGCTAAATTTCCA	
tcdA	NK1	PCR primer	GGACATGGTAAAGATGAATTC	
NK2	PCR primer	CCCAATAGAAGATTCAATATTAAGCTT	

LAMP reaction

The LAMP reactions were performed in a 25 μl reaction mixture (DNA amplification kit; Eiken Chemical Co., Ltd., Tochigi, Japan) containing the following reagents in the final concentration: 20 mM Tris–HCl (pH 8.8), 10 mM (NH4) 2SO4, 10 mM KCl, 0.1% Tween-20, 0.8 M betaine, 8 mM MgSO4, 1.4 mM deoxynucleoside triphosphate and 8 U Bst DNA polymerase. Each LAMP reaction, using a real-time turbidimeter, was composed of 40 pmol FIP and BIP, 20 pmol LB and LF, 5 pmol F3 and B3 primers, and 2 μl DNA template. An additional 1 μl of calcein/Mn2+ complex (Fluorescent Detection Reagent; Eiken Chemical Co., Ltd., Tochigi, Japan) was added if direct visual inspection was required. The reaction was conducted in reaction tubes (Eiken Chemical Co., Ltd., Tochigi, Japan) within 60 min at an isothermal temperature of 66 °C.

Detection of LAMP products

Data were collected as previously described in Lin et al. (2022). Specifically, two different methods, chromogenic reaction with calcein/Mn2+ complex and real-time monitoring of turbidity, were applied to detect LAMP products. For direct visual inspection, 1 μl of calcein (fluorescent detection reagent; Eiken Chemical Co., Ltd., Tochigi, Japan) was added to 25 μl of reaction mixture in a LAMP tube before the LAMP reaction. For a positive reaction, the color changed from orange to green, while a negative reaction failed to turn green and remained orange. The color change could be observed by naked eye observation under natural light or 365 nm ultraviolet light. For assessing turbidity (Mori et al., 2001), real-time amplification was monitored through spectrophotometric analysis by measuring the optical density (λ650 nm) at 400 nm every 6 s with the aid of a Loopamp real-time turbidimeter (LA-230; Eiken Chemical Co., Ltd., Tochigi, Japan).

PCR detection

The PCR conditions used for amplification were described previously (Kato et al., 1991).

Electrophoresis using a 2% agarose gel (Amresco, Solon, OH, USA) containing ethidium bromide was applied to analyze the PCR-amplified products. Images were captured using a Bio-Rad Gel Doc EQ Imaging System (Bio-Rad, Hercules, CA, USA).

Statistical analysis

The required sample size was estimated using Buderer’s method (Buderer, 1996), setting the Z-value at 1.96 for the normal distribution and constraining the width of the 95% confidence interval to a maximum of 10%. Previous similar research reported a specificity and sensitivity of 95% for the LAMP assay (Soroka, Wasowicz & Rymaszewska, 2021). Given the prevalence of CDI in China is 11% (Wen et al., 2023), the sample size of the study had to be at least 160 to ensure statistical validity. The McNemar test was used to analyze count data, and the Cohen’s kappa (κ) statistic was employed to evaluate the agreement between the LAMP and PCR methodologies. A κ correlation value of 0.40 or below signifies a weak level of agreement, a value ranging from 0.41 to 0.60 reflects moderate agreement, and a value exceeding 0.60 denotes a strong agreement between observations. The specificity, sensitivity, positive predictive value (PPV), and negative predictive value (NPV) of the LAMP and PCR methods were calculated using standard formulas and then compared to results obtained from gene sequencing. Each metric was assessed by analyzing the agreement between the LAMP or PCR results and the gene sequencing data. All statistical analyses were performed using SPSS software version 26.0 (IBM Corp., Armonk, NY, USA). A p-value of <0.05 was considered statistically significant.

Results

Optimal primers for rapid detection of tcdA

All three sets of primers designed herein (Table 2) produced turbidity after 26 min (Fig. 1). Primer set tcdA12 showed the fastest amplification (Fig. 1), so tcdA12 primers were selected for optimization. The reaction time of tcdA12 primers with additional loop primers (LB and LF) was less than one-half that of the tcdA8 primers without loop primers.

Figure 1 Three sets of primers amplified tcdA by measuring the optical density using a Loopamp real-time turbidimeter at 650 nm every 6 s.

The tcdA12 primer set was applied with loop primers, whereas the remaining sets were applied without loop primers.

Appropriate temperature for tcdA LAMP reaction

The tcdA12 primer set was evaluated across a temperature range of 58 °C to 69 °C at intervals of 1 °C. The ideal temperature range for the tcdA12 primer set was determined to be 60 °C to 67 °C, with peak amplification efficiency identified at 66 °C (Fig. 2).

Figure 2 Temperatures from 58 °C to 69 °C, at 1 °C intervals, were observed to confirm the most appropriate temperature for the tcdA LAMP reaction. Turbidity was monitored by a Loopamp real-time turbidimeter at 650 nm every 6 s.

Specificity of tcdA LAMP reaction

Primer set tcdA12 only amplified when C. difficile VPI10463 was used as the template; all reactions with other bacterial species and no template controls were negative (Fig. 3A). Results of the chromogenic reaction confirmed from the results of the real-time monitoring of turbidity (Fig 3B).

Figure 3 Specificity of the LAMP reaction for the detection of tcdA.

(A) Turbidity was monitored by a Loopamp real-time turbidimeter at 650 nm every 6 s; (B) a visual inspection method was compared. Two reactions were performed at 66 °C for 60 min. Lines: 1, Acinetobater baumannii H18; 2, Betahaemolytic streptococcus group A CMCC 32213; 3, Bordetella pertussis ATCC 18530; 4, Bacillus megatherium 4623; 5, Bacillus anthracis ATCC 9372; 6, Corynebacterium diphtheria CMCC 38001; 7, Enteropathogenic E. coli 2348; 8, Enterotoxigenic E. coli 44824; 9, Enteroinvasive E.coli 44825; 10, Mycobacterium tuberculosis 8362; 11, Neisseria meningitides group B CMCC29022; 12, Pseudomonas maltophilia ATCC13637; 13, Pseudomonas aeruginosa CMCC 10104; 14, Shigella flexneri 4536; 15, Shigella sonnei 2531; 16, Salmonella 10025819551001; 17, Salmonella paratyphosa 86423; 18, Salmonella aberdeen 9264; 19, Salmonella enteritidis 50326; 20, Staphylococcus aureus 2740; 21, Stenotrophomonas maltophilia H62; 22, Vibrio parahaemolyticus 5474; 23, Vibrio cholera 3802; 24, Vibrio carchariae 5732; 25, Yersinia enterocolitica 1836; 26, Yersinia pestis 2638; 27, Positive control (C. difficile VPI10463); 28, Negative control (distilled water).

Sensitivity of tcdA LAMP reaction

As demonstrated in Figs. 4A and 4B, the detection limit of real-time turbidity was 20.700 pg/µl, which was identical to that of visual detection. The NK1 and NK2 primers with the same concentration of C.difficile VPI10463 DNA were also evaluated using PCR. The detection limit for PCR was 207 pg/µl (Fig. 4C), which was 10-fold lower than that of the LAMP reaction.

Figure 4 Comparison of sensitivity of LAMP and PCR for the detection of tcdA.

The pure genomic DNA extracted from C. difficile VPI10463 was diluted 10-fold from 207 ng/µl to 0.000207 pg/µl. Both LAMP (A, B) and PCR (C) were conducted in duplicate for each dilution point. The two LAMP reactions (A, B) were performed at 66 °C for 60 min. (A) Turbidity was monitored by a Loopamp real-time turbidimeter at 650 nm every 6 s; (B) The visual colour detection was compared using the addition of 1 µl of calcein/Mn2+ complex to 25 µl of the LAMP reaction mixture before the LAMP reaction; (C) PCR products were analyzed by 2% agarose gel electrophoresis and stained with ethidium bromide. Tubes and lanes: 1, 207 ng/µl; 2, 20.7 ng/µl; 3, 2.07 ng/µl; 4, 207 pg/µl; 5, 20.7 pg/µl; 6, 2.07 pg/µl; 7, 0.207 pg/µl; 8, 0.0207 pg/µl; 9, 0.00207 pg/µl; 10, 0.000207 pg/µl; 11, Negative control (distilled water); M, D2000 DNA Marker (Tiangen Biotech Co., Ltd.).

Evaluation of the assay with stool specimens

A total of 164 stool specimens were considered eligible and suitable for this study. One patient was diagnosed with pseudomembranous enteritis which is a common symptom of CDI (Figs. 5A, 5B). Electronic colonoscopy revealed numerous scattered pale yellow pseudomembranes and areas of congested, brittle mucosa (Fig. 5A). Histopathological imaging (Fig. 5B) revealed an infiltration of inflammatory cells into the mucosal lamina propria.

Figure 5 Electronic colonoscopy and histopathological photographs of pseudomembranous colitis.

(A) Electronic colonoscopy of pseudomembranous enteritis (60×). Electronic colonoscopy revealed scattered pale yellow pseudomembranes (blue arrows) and congested, brittle mucosa (black arrows); (B) histopathologic photographs of pseudomembranous colitis (100×). The blue arrows indicated the infiltration of inflammatory cells into the mucosal lamina propria.

In the stool specimens, the detection rate of the tcdA gene using the LAMP method was 17% (28/164), significantly higher than the 10% (16/164) detection rate of the PCR method (X2 = 47, df = 1, P < 0.01) (Table 3). The consistency between the LAMP and PCR methods was moderate (Kappa = 0.533, p < 0.01). Notably, 15 of the stool specimens that were negative for the tcdA gene in PCR but positive in LAMP were subsequently confirmed to be positive through Sanger sequencing performed at Sangon Biotech (Shanghai, China). Of the three specimens that were negative in LAMP but positive in PCR, only one was found to be positive using tcdA gene sequencing. Using sequencing as the reference standard, the LAMP assay outperformed the PCR assay with a sensitivity of 97% compared to 48%, and a specificity of 100%, nearly identical to PCR’s 99%. The LAMP assay achieved a PPV of 100% and an NPV of 99%, whereas the PCR assay had a PPV of 88% and an NPV of 90% (Tables 4 and 5).

Table 3 Comparison of the results of LAMP and PCR for detecting the tcdA gene in stool specimens.

PCR	LAMP	Total	
Positive	Negative	
Positive	13	3	16	
Negative	15	133	148	
Total	28	136	164	

Table 4 Comparison of the results of LAMP and sequencing for detecting the tcdA gene in stool specimens.

Sequencing	LAMP	Total	
Positive	Negative	
Positive	28	1	29	
Negative	0	135	135	
Total	28	136	164	

Table 5 Comparison of the results of PCR and sequencing for detecting the tcdA gene in stool specimens.

Sequencing	PCR	Total	
Positive	Negative	
Positive	14	15	29	
Negative	2	133	135	
Total	16	148	164	

Discussion

This study marks a significant advancement in the rapid detection of C. difficile. The optimized LAMP assay reduced the time to detect the tcdA gene to under 60 min. Incorporating loop primers (LB and LF) reduced LAMP reaction times by over 50%. Including loop primers also enhanced the LAMP reaction’s efficiency and sensitivity by multiplying the initiation sites for amplification. Loop primers are designed to be activated singularly during the synthesis of the artificial template (Nagamine, Hase & Notomi, 2002). LAMP-based detection yielded negative results for 26 pathogenic bacterial strains from different genera than C. difficile, demonstrating the primers’ exceptional specificity. The 10-fold higher sensitivity of the LAMP method compared to conventional PCR corroborates findings from prior studies (Kim et al., 2022a; Carvajal-Gamez et al., 2023) and supports the integration of this LAMP assay into routine diagnostic workflows.

The LAMP method yielded a detection rate of 17% for the tcdA gene of C. difficile in stool specimens, which is similar to rates observed in diarrheal stool samples of diverse populations across China. For instance, a multicenter study in Shanghai, China, reported a detection rate of 18% (93/531) for the tcdA gene (Mi et al., 2020), while an independent cross-sectional study in Southwest China noted a detection rate of 14% (125/978; Liao et al., 2018). These findings suggest that the LAMP method’s efficacy in detecting the tcdA gene is comparable to the methods used in previous studies. Moreover, this study also revealed a significant discrepancy in the detection rates of the tcdA gene in stool specimens between LAMP and PCR. LAMP showed a superior detection rate. This was further evidenced by the fact that 15 of the stool specimens that were negative in PCR, but positive in LAMP, were subsequently confirmed to be true positives through sequencing of the tcdA gene. In our previous study of 300 cultured C. difficile strains from Southern Medical University Nanfang Hospital, a perfect concordance (kappa = 1) was observed between LAMP and PCR detections of the tcdA gene (Lin et al., 2022), supporting the notion that LAMP is equally reliable when bacterial DNA is present in higher quantities. The difference in consistency between LAMP and PCR in detecting C. difficile strains in stool specimens in this study may be attributed to the variable concentration of the C. difficile DNA present. In stool specimens where C. difficile DNA may be at relatively low concentrations, LAMP’s lower detection limit allows for a higher detection rate. Conversely, the increased DNA content in cultured C. difficile strains following enrichment culture may mask the differences in detection limits between the two methods, leading to a high degree of consistency in detecting the tcdA gene in these strains. These findings collectively advocate for the integration of LAMP in clinical diagnostics of CDI, particularly for cases where PCR may not provide sufficient sensitivity.

In the analysis of stool samples, the LAMP assay showed high sensitivity (97%) and perfect specificity (100%), along with a PPV of 100% and an NPV of 99%, when using sequencing as the gold standard. This suggests that the LAMP assay offers a high degree of reliability in both detecting the presence of the target pathogen and verifying its absence. In contrast, the PCR assay, despite its high specificity (99%), demonstrated significantly reduced sensitivity (48%), potentially limiting its effectiveness as a standalone diagnostic tool. The PCR’s PPV of 88% and NPV of 90% suggest that while positive results are likely accurate, the risk of false negatives is increased, which could contribute to under-diagnosis. Consequently, the LAMP assay could represent a more robust approach for precise detection of CDI within the examined population.

The LAMP method exhibits considerable advantages in the detection of the tcdA gene for C. difficile. First, eliminating the DNA denaturation step simplifies the operational workflow, as reactions are carried out at an isothermal condition of 60–65 °C using the strand displacement activity of Bst DNA polymerase, thus obviating the need for the precise thermal cycling equipment required for PCR (Soroka, Wasowicz & Rymaszewska, 2021). This significantly lowers the technical requirements of detection, allowing molecular diagnostic techniques to be performed in resource-limited settings. Second, the LAMP reactions can be directly visualized through color changes in the calcium-magnesium complex when exposed to UV light, circumventing the need for the complex gel electrophoresis step that is requisite in PCR. This advancement not only expedites detection speed, but also minimizes the dependency on carcinogenic dyes, thereby enhancing the safety profile of the procedure. Third, the sensitivity of LAMP is 10–100 times higher than that of conventional PCR (Kim et al., 2022a; Ashmi et al., 2023; Carvajal-Gamez et al., 2023). This significant enhancement in sensitivity ensures accurate detection of pathogens even at low concentrations, which is vital for early intervention and for controlling the spread of infection. Finally, the specificity of LAMP is notably enhanced through the employment of 4–6 primers, which identify up to eight specific locations on the DNA template. This significantly mitigates the likelihood of false positives, a potential risk associated with the two-primer system in PCR (Soroka, Wasowicz & Rymaszewska, 2021).

While the LAMP method stands out for its advantages, it is not without its limitations. Cross-contamination arising from multiple pipetting steps is a notable concern, particularly with material present in the aerosol. This study mitigated this issue by employing low-melting-point paraffin to seal the reaction mixtures, thereby preventing the spread of amplification products. Rigorous sample handling protocols and enhanced ventilation should be conducted to minimize contamination risks. Another limitation is the suitability of LAMP products for downstream applications, such as sequencing or cloning (Sahoo et al., 2016). This drawback underscores the need for further methodological refinements to expand the utility of LAMP products beyond mere detection. Additionally, the sensitivity of LAMP to inhibitors remains a complex challenge (Dong et al., 2014). This highlights a potential area for the development of more sophisticated controls and detection metrics within the LAMP protocol. Finally, primer-dimer formation in LAMP could lead to false positives.

A limitation of this study is the exclusive use of the C. difficile VPI10463 strain as a positive control, which may not represent the diversity of C. difficile strains. To address this limitation and to understand its potential impact on the conclusions drawn from the results, a more comprehensive analysis was performed. The subsequent sequencing analysis of 15 stool specimens that tested positive with the LAMP assay but negative with PCR provided additional insights. Not only did this analysis confirm the presence of C. difficile VPI10463 in two cases, but it also identified other strains in the remaining samples: seven cases of GZ14, two cases of ZR48, and one case each of ZR80, ZR50, GZ5, and SH8. These findings suggest that the LAMP assay developed in this study may have the capacity to detect a broader spectrum of C. difficile strains than initially anticipated, which is a significant consideration for the validity of this method. This understanding of the assay’s limitations and their possible impact on the conclusions drawn in this study is critical for interpreting the study’s results and for guiding future research directions.

Conclusions

In this study, the LAMP method proved to be an effective technique for the rapid and visual detection of the tcdA gene of C. difficile, demonstrating potential advantages over PCR in terms of speed, simplicity, and sensitivity. The tcdA-LAMP assay is particularly suitable for medical diagnostic environments with limited resources and represents a promising diagnostic strategy for the screening and detection of CDI in populations at high risk.

Supplemental Information

Supplemental Information 1 Sequence comparison of tcdA gene carried by Clostridioides difficile VPI10463 with the reported tcdA gene in GenBank.

Supplemental Information 2 STROBE checklist.

Supplemental Information 3 Raw data.

We would like to thank the Institute of Disease Control and Prevention and the Academy of Military Medical Sciences for their technical assistance.

Additional Information and Declarations

Competing Interests

Author Contributions

Human Ethics

DNA Deposition

Data Availability

The authors declare that they have no competing interests.

Minyi Lin conceived and designed the experiments, performed the experiments, analyzed the data, prepared figures and/or tables, authored or reviewed drafts of the article, and approved the final draft.

Pu Wang conceived and designed the experiments, performed the experiments, analyzed the data, prepared figures and/or tables, authored or reviewed drafts of the article, funding Acquisition, and approved the final draft.

Bingyun Lu conceived and designed the experiments, performed the experiments, analyzed the data, prepared figures and/or tables, and approved the final draft.

Ming Jin conceived and designed the experiments, performed the experiments, prepared figures and/or tables, and approved the final draft.

Jiasheng Tan conceived and designed the experiments, performed the experiments, prepared figures and/or tables, and approved the final draft.

Wei Liu conceived and designed the experiments, performed the experiments, prepared figures and/or tables, technical support, and approved the final draft.

Jing Yuan conceived and designed the experiments, performed the experiments, prepared figures and/or tables, technical support, and approved the final draft.

Xiaomou Peng conceived and designed the experiments, analyzed the data, authored or reviewed drafts of the article, project Administration, and approved the final draft.

Ye Chen conceived and designed the experiments, analyzed the data, authored or reviewed drafts of the article, funding Acquisition; Project Administration, and approved the final draft.

The following information was supplied relating to ethical approvals (i.e., approving body and any reference numbers):

Fudan University Affiliated Huashan Hospital granted Ethical approval to carry out the study within its facilities (Ethical Application Ref: FDEC-2012-014).

The following information was supplied regarding the deposition of DNA sequences:

The tcdA gene sequences of C. difficile are available at GenBank: X92982.1.

https://www.ncbi.nlm.nih.gov/nuccore/X92982.1/

The following information was supplied regarding data availability:

The sequence comparison of tcdA gene carried by Clostridioides difficile VPI10463 with the reported tcdA gene in GenBank are available in the Supplemental Files. The raw data includes Figs. 1, 2, 3A, 3B, 4A–4C, and Table 3.

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
