# Peer review of "Development and evaluation of a rapid visual loop-mediated isothermal amplification assay for the tcdA gene in Clostridioides difficile detection"

_PeerJ, doi:10.7717/peerj.17776_

## Round 0.1 · original submission · Major Revisions

These suggested revisions are a guide and are not comprehensive. I stopped at line 261, as reviewer 1 called for a complete rewrite of Discussion.
Line 50. Here and throughout, use an appropriate number of significant figures for the resolution of the measurement (21 pg/ul). See also line 67 (144 not 143.6).
Line 66. “in 10 United States sites” is vague
Line 71. “are thought to be 2” or “are” Do you have reason to doubt Kuehne et al.? Also “two” not “2”
Line 78. Provide reference for “a study”
Line 81. Italicize “in vitro” and “in vivo”
Line 82. “Besides, either one of toxin A and toxin B is produced,” “What’s more..” is awkward.
Line 85. Delete “Therefore…
Line 99. Do not say “reported to be” unless you doubt those reports.
Line 105-106. “It relies on the strand displacement activity of the Bst DNA
106 polymerase under isothermal conditions within 60 minutes” Replace with “LAMP uses… “ Also, LAMP doesn’t rely on “within 60 minutes.” Delete that phrase.
Line 108. Avoid starting sentences with “it”
Line 111. “by naked eyes instead of complex instruments” is awkward.
Line 115. Replace “This study aimed…original” with “In this study we designed four novel…”
Line 116-118. Delete phrase “Our data analysis..” with something specific (comparison of ## strains…). Also, “the” implies one of the four primers worked. State that. Finally, state if LAMP and PCR agreed.
Line 161. “According to…” suggests the sequence was a manual. State that this sequence was selected as a reference for primer design.
Line 197. Delete “those as”
Line 198. Do not start a sentence with a number.
Line 199. Images are captured not “reported”
Line 200. Do you mean “PCR experiments were repeated twice” for “results were double-checked”?
Line 204, “Calculated…” is not a complete sentence.
Line 206-208. Belongs in Discussion.
Line 217. Replace with “All four sets of primers designed herein (Table 2) produce turbidity within 26 min (Fig. #).”
Line 219. Move interpretation “indicating..” to Discussion. Also, revise to “Primer set tcdA12 primers showed the fastest amplification (Fig, 1) and were selected for optimization”
Line 221. Delete “From our data, it was shown that.” By definition, results presents your data.
Line 223. I don’t understand how your amplification with novel primers could be in “accordance with a previous report..” Move to Discussion.
Line 227. Delete “In order” and move methods to Methods.
Line 229. Take out phrases like “As shown in Figure 2.” State result and then direct reader to the figure or table.
Line 230-231. Wordy.
Line 258-261. Move to Methods.
Line 378. Be consistent in references. All previous journal titles were all caps.
Line 403. All previous titles only capped proper nouns. (Not Burden, Infection and Outcomes)
Line 407. Italicize genus and species.
Resolution of figures is poor and horizontal lines add little.
Figure 3 needs revision it should show two lines and indicate what they are. Also, differences between positive and negative tubes in Fig. 3 B is not clear. Could you try a white background? Figure 6B is better though.

Reviewer 1 ·

Basic reporting

1. The manuscript must be revised and grammatical errors should be corrected.
2. More reviews should be included in the manuscript
3. At this stage, the discussion part is not at the publication stage. There is a repetition of result finding in the discussion. It should be analytical, concise, and more reviews to support the findings.
The conclusion part should be rewritten again.
4. In Fig 1, A1 primers did not show any turbidity. It needs to be addressed properly.
5. Fig 3B, The figure should be modified. Positive and negative result should be indicated in symbol (+) and (-) respectively in front of the tube.
6. In Fig 4B, Positive and negative result should also be indicated in symbol (+) and (-) respectively in front of the tubes.
7. Fig 4C should be revised again
8. In Fig 5A, arrow mark (→) should be included to indicate scattered pale yellow microscopic particles and congested mucosa
9. In Fig 5B. arrow mark (→) should be included to indicate neutrophils, lamina propria, and epithelial cells
10. in Line L 1 Title of this manuscript should be modified and should be concise. ‘Clinical application’ may be modified to ‘evaluation’
11. L 126 sentence needs to be rewritten.
12. L 157 Mention the concentration of genomic DNA extracted from stool instead of volume
13. L 203-L 213 Modify the sentences. Not looking good.

Experimental design

1. Conventional PCR is not the gold standard test for the detection of Clostridium difficile in stool samples. Different tests are available for the detection of toxins of Clostridium difficile. So it needs to be addressed properly with the earlier report
2. In this study, only the detection limit of LAMP was mentioned properly and a comparison regarding the detection limit between PCR and LAMP was given. There is a lack of specificity. Testing with other bacteria was done adequately, but checking with other strains of Clostridium difficile other than C. difficile VPI10463 is lacking. It should be done.
3. Statistical analysis should be performed again. The sensitivity and specificity of the LAMP assay for stool samples were not provided properly. Here authors mentioned only the detection rate which is also too low i. e 17.1%

Validity of the findings

no comment

Additional comments

In this study, Lin et al. developed Loop mediated amplification assay, utilizing four sets of primers, targeting tcdA gene for rapid and visual detection of Clostridium difficile in stool samples. They reported the detection limit of this assay of 20.7 pg/µl.

This is an interesting study and the LAMP assay developed could potentially be a valuable diagnostic test to detect Clostridium difficile.

Reviewer 2 ·

Basic reporting

The article is well-written. The English is understandable.

Unfortunately, the References section needs improvement; most of the publications are outdated, some even from the 90s. It is necessary to remove old references and include new ones from the last 3-4 years.

Experimental design

The methodology is correct and includes details such as the used reagents, their concentrations, names of manufacturers, etc. The aim was presented and achieved.

The authors state that all LAMP and PCR results were double-checked. Typically, experiments are conducted in three repetitions. In these studies, it is unclear whether double-checking sufficed or if the results from both repetitions were similar enough to obviate the need for a third repetition.

Validity of the findings

A rapid and cost-effective detection method for the tcdA gene in Clostridioides difficile (C. difficile) is crucial due to its significance in identifying toxin-carrying strains. In the reviewed paper, a specific and sensitive loop-mediated isothermal amplification (LAMP) assay was developed and validated. The LAMP assay successfully amplified and visualized the target DNA within 60 minutes at 66°C, exhibiting higher specificity than conventional PCR. The detection limit of LAMP was 10 times more sensitive than PCR, and the method demonstrated a significantly higher detection rate (17.1%) in 164 stool specimens compared to PCR (9.8%). The LAMP assay proves advantageous for rapid, simple, and visual detection of the tcdA gene in C. difficile strains and stool specimens.

---

## Round 0.2 · Minor Revisions

The science appears sound but one reviewer commented that the Introduction needs revision and the text should be checked for grammatical errors throughout. If those concerns are addressed, I do not see a need for another round of peer review.

Regards,

Michael

**Language Note:** The Academic Editor has identified that the English language must be improved. PeerJ can provide language editing services - please contact us at [email protected] for pricing (be sure to provide your manuscript number and title). Alternatively, you should make your own arrangements to improve the language quality and provide details in your response letter. – PeerJ Staff

Reviewer 1 ·

Basic reporting

1. Authors have improved the manuscript and all comments have been readdressed properly but still some revisions are required to make this manuscript into publication standards.
2. Introduction part is not at the publication stage. It should be rewritten properly
3. Grammatical errors should be rectified through the manuscript

Experimental design

Although sequencing of other strains than C. difficile VPI10463 was done but checking with LAMP assay and PCR was missing. It needs to be addressed properly.

Validity of the findings

Statistical analysis results should be rechecked

Additional comments

1. L 121 sentence should be rewritten.
2. L126-L130 Not looking good. Sentences should be rewritten.
3. L259-L260 Modify sentences.
4. L297 Not looking good. Sentence should be rewritten
5. L332-L342 Modify sentences. Not looking good
6. L343-363 Modify sentences. Not looking good

Reviewer 2 ·

Basic reporting

The authors significantly corrected the manuscript according to the reviewer's suggestions. Recently, I recommend the article for publication.

Experimental design

The authors significantly corrected the manuscript according to the reviewer's suggestions. Recently, I recommend the article for publication.

Validity of the findings

The authors significantly corrected the manuscript according to the reviewer's suggestions. Recently, I recommend the article for publication.

---

## Round 0.3 · Minor Revisions

Both reviewers found the paper acceptable but I still have a number of comments that should be considered. Importantly, I don't understand how DNA sequencing was used as the gold standard for identifying CDI-positive stool samples. My specific comments are indicated in the attached pdf.

Michael

Reviewer 1 ·

Basic reporting

The authors have sufficiently improved the manuscript. Recommended for publication

Experimental design

Nothing to be revised

Validity of the findings

Nothing to be verified

---

## Round 0.4 · Minor Revisions

I found several issues of format and style that must be addressed. I am not sure if I missed them in a previous version or my comments were lost in revisions. To ensure timely processing of this important manuscript document every revision. Specifically, I made 21 comments in the attached pdf. (..v3_MgL.pdf). Include a document where you list these comments and indicate how, by line number, the comment was addressed. Do not simply state "all suggested changes were made."

---

## Round 0.5 · accepted · Accept

I have two minor changes that can be addressed in production. In Abstract, you need to define the primers you used as follows.
"the target DNA was successfully amplified with a set of primers designated tcdA12..."
Also, revise line 59 and subsequent text to a reasonable number of significant figures as follows "The detection limit of LAMP was 21 pg/µl..."

Regards,

Michael